# Effect of Dietary Supplementation of *Lactiplantibacillus plantarum* N-1 and Its Synergies with Oligomeric Isomaltose on the Growth Performance and Meat Quality in Hu Sheep

**DOI:** 10.3390/foods12091858

**Published:** 2023-04-29

**Authors:** Zhiqiang Zhou, Xinyi Xu, Dongmei Luo, Zhiwei Zhou, Senlin Zhang, Ruipeng He, Tianwu An, Qun Sun

**Affiliations:** 1College of Biomass Science and Engineering, Sichuan University, Chengdu 610064, China; 2Key Laboratory of Bio-Resources and Eco-Environment of the Ministry of Education, College of Life Sciences, Sichuan University, Chengdu 610064, China; 3Sichuan Academy of Grassland Sciences, Chengdu 611731, China

**Keywords:** probiotics, synbiotics, growth performance, meat quality, flavor, texture

## Abstract

Probiotics have gained tremendous attention as an alternative to antibiotics, while synbiotics may exhibit a greater growth promoting effect than their counterpart probiotics due to the prebiotics’ promotion on the growth and reproduction of probiotics. The objective of this study was to investigate the influence of *Lactiplantibacillus plantarum* N-1 and its synbiotic with oligomeric isomaltose on the growth performance and meat quality of Hu sheep. Hu sheep (0–3 days old) were fed with water, probiotics of N-1, or synbiotics (N-1 and oligomeric isomaltose) daily in three pens for 60 days and regularly evaluated to measure growth performance and collect serum (five lambs per group). *Longissimus thoracis* (LT) and *biceps brachii* (BB) muscle tissues were collected for the analysis of pH value, color, texture, nutrients, mineral elements, amino acids, volatile compounds, and antioxidant capacity. The results showed that dietary supplementation of N-1 tended to improve growth performance and meat quality of Hu sheep, while the synergism of N-1 with oligomeric isomaltose significantly improved their growth performance and meat quality (*p* < 0.05). Both the dietary supplementation of N-1 and synbiotics (*p* < 0.05) increased the body weight and body size of Hu sheep. Synbiotic treatment reduced serum cholesterol and improved LT fat content by increasing the transcription level of fatty acid synthase to enhance fat deposition in LT, as determined via RT-qPCR analysis. Moreover, synbiotics increased zinc content and improved LT tenderness by decreasing shear force and significantly increased the levels of certain essential (Thr, Phe, and Met) and non-essential (Asp, Ser, and Tyr) amino acids of LT (*p* < 0.05). Additionally, synbiotics inhibited the production of carbonyl groups and TBARS in LT and thus maintained antioxidant stability. In conclusion, it is recommended that the use of synbiotics in livestock breeding be promoted to improve sheep production and meat quality.

## 1. Introduction

With the growing improvement in people’s living standards, meat consumption is also increasing. At the same time, consumers also pay more attention to the quality of meat products. Lamb is popular among consumers because of its juiciness, unique flavor, high protein, low cholesterol, and essential amino acids and fatty acids [1]. However, the consumption of lamb is usually limited by its odor and short shelf life [2]. Meanwhile, the improper use of antibiotics in sheep farming may cause drug residues in meat products [3], which can be harmful to humans. Therefore, to meet the increasing demand for high-quality lamb, improving meat quality has become a hot topic in the field of animal nutrition. The Hu sheep is a first-class protected local livestock breed in China, with excellent traits such as two litters a year, multiple lambs per litter, good lactation performance, fast growth and development, ideal meat production performance after improvement, and resistance to high temperature and humidity [4].

Recently, probiotics have received significant attention as an alternative to antibiotics [5,6]. Previous studies have shown that probiotics have the potential to improve microbial balance and the intestinal environment [7]. Prebiotics are organic substances that cannot be digested and absorbed by the host but can selectively promote the metabolism and proliferation of beneficial bacteria in the body, thereby improving the health of the host [8]. Available studies have reported that probiotics and synbiotics can improve the growth performance and meat quality of sheep through various pathways [8,9]. One study by Saleem [10] showed that treatment with probiotics (*Pediococcus acidilactici* and *Pediococcus pentosaceus*) improved animal health and nutrient digestibility in postweaning lambs. A recent study reported that *Lactobacillus casei* HM-09 and *Lactobacillus plantarum* HM-10 treatment improved the tenderness and flavor of lambs [6]. In addition, probiotics may be beneficial in increasing the number of cellulolytic bacteria in the rumen of ruminants [11], thereby increasing fiber digestibility, improving nutrient synthesis and bioavailability, and contributing to improved growth performance [12]. The supplementation of diets with probiotics has been shown to improve growth performance in monogastric animals, such as broilers [13] and piglets [14], with comparable or better performance compared to that achieved with antibiotics [15]. Furthermore, prebiotics are essential for the better survival of probiotics in the intestinal tract. With the assistance of prebiotics, probiotics can live well in the digestive system and tolerate anaerobic environments well, such as low oxygen, low temperature, and pH [16].

The aforementioned studies have shown the potential of probiotics or synbiotics as feed additives to improve antioxidant and meat quality in livestock and poultry. Moreover, there is increasing interest in research on their application in ruminant production, as probiotics help maintain the balance of the gut microbiota and can be a possible alternative to the use of traditional antibiotics. Our previous study [17] showed that *Lactiplantibacillus plantarum* N-1 (a strain isolated from traditional cheese in the previous study) has probiotic properties such as antioxidant capacity, adhesion to Caco-2 cells, and sensitivity to antibiotics in vitro. Further, *L. plantarum* N-1 was able to improve lipid metabolism in rats by promoting the production of short-chain fatty acids in the intestine of hypercholesterolemic rats and inhibiting the expression of HMG-CoA reductase [17]. The objective of this study was to investigate potential feed additives for improving the growth performance and meat quality of sheep and to further explore effective ways to improve the meat quality of lambs. Currently, there is a lack of probiotics that can replace antibiotics in the farming industry. Consequently, in order to explore the potential probiotic effect of *L. plantarum* N-1, we selected *L. plantarum* N-1 for further investigation. In addition, we used oligomeric isomaltose in combination with *L. plantarum* N-1 and explored the mechanism of improving the growth performance of lambs from intramuscular fat deposition of lambs and evaluated the potential of probiotics and synbiotics to improve lamb meat quality.

## 2. Materials and Methods

### 2.1. Experimental Design and Sample Collection

This research was performed on a sheep farm (longitude 104°85′ E, latitude 29°76′ N) in Neijiang City, Sichuan Province, China, from January to March 2022. The average temperature during the experimental period was 14 °C, with a minimum of 11 °C and a maximum of 19 °C. The Hu sheep were farm-born and reared with their dams until the end of the experiment. Fifteen male lambs (initial body weight 4.72 ± 0.61 kg) were randomly assigned to the control, probiotics treatment group, and synbiotics treatment group according to their weight. Each group (five lambs and five ewes) was kept in separate pens (pen size 4 m × 4 m, fence height 0.8 m), and lambs were fed by breastfeeding until the end of the experiment. Additionally, based on the daily basal diet, the probiotics group was fed (by gavage) *Lactiplantibacillus plantarum* N-1 (1 × 10^9^ CFU/g per kilogram body weight), the synbiotics group was fed *Lactiplantibacillus plantarum* N-1 (1 × 10^9^ CFU/g per kilogram body weight) and oligomeric isomaltose (0.08 g/kg body weight), and the control group was fed the same amount of water. The study began 3 days after the lambs were acclimated to the experimental environment and lasted for 60 days. During the research, the weight and body size of lambs were measured (animal weight was measured by farm staff, and the length, height and chest circumference were determined by trained sheep farm staff using measuring tape) every 15 days, and serum samples were collected every month. All procedures were approved by the Ethics Committee of the College of Life Sciences, Sichuan University (approval number SCU230319001).

### 2.2. Sample Collection

At the conclusion of the study, all lambs were transported to a nearby commercial abattoir and humanely slaughtered in accordance with animal welfare procedures. Prior to slaughter, the lambs underwent a 12 h fast from both solids and liquids. Then, the *longissimus thoracis* (LT) and *biceps brachii* (BB) of the left hind leg of the carcasses were collected. All samples were stored in ice boxes and quickly returned to the laboratory for storage at 4 °C and −20 °C for subsequent analysis.

### 2.3. Determination of Blood Lipids in Hu Sheep Serum

Total cholesterol, triglyceride, low-density lipoprotein cholesterol, and high-density lipoprotein cholesterol were determined using a commercially available assay kit (Nanjing Jiancheng Institute of Biological Engineering Co., Nanjing, China).

### 2.4. Meat Quality Analysis

Meat quality was determined using the *longissimus thoracis* (LT) and *biceps brachii* (BB) muscle samples. The pH values at 1 h and 24 h after slaughter were measured by a portable pH meter (POCKET pH TESTER, Thermo Fisher Scientific Inc, Waltham, MA, United States). To determine the color of Hu sheep tissue, a colorimeter (CM-2300d, Konica Minolta Investment Ltd., Tokyo, Japan) with illuminant D65 and 10° standard observer was used. Each meat sample was placed vertically at the measurement probe and fastened during the test, with measurements of each sample repeated three times, and the luminance (*L**), redness (*a**), and yellowness (*b**) were then measured.

Briefly, fat and fascia were removed from LT and BB tissues and cut into strips of meat (3 × 1 × 1 cm^3^), which were then weighed and recorded as *m_1_*. Next, the meat strips were placed in a water bath at 80 °C for 30 min and stirred every 10 min. Once finished, any surface water was wiped away and the weight of the Hu sheep tissue was measured again and recorded as *m_2_*. The cooking loss of meat was calculated as follows: cooking loss (%) = (*m_1_* − *m_2_*)/*m_1_* × 100%. Meanwhile, after 48 h of hanging at 4 °C, the surface water of tissues was wiped off and the weight of the meat was recorded as *m_3_*. The dripping loss was calculated as follows: dripping loss (%) = (*m_1_* − *m_3_*)/*m_1_* × 100%. In addition, the water holding capacity was measured using the centrifugation method (4000× *g* for 30 min at 4 °C).

### 2.5. Determination of the Chemical Composition of Hu Sheep Muscle

In brief, the crude fat, crude protein, moisture and ash content of LT tissues were determined based on the Association of Official Analytical Chemists (AOAC) [18].

### 2.6. Determination of the Mineral Elements in Hu Sheep Tissue

Ten mineral elements (K, Na, Ca, Mg, Fe, Zn, Cu, Cr, Se, and Mn) in LT muscle were measured using an inductively coupled plasma mass spectrometer (7900 ICP-MS, Agilent Technologies (China) Ltd., Beijing, China) after digestion with nitric acid. The determination was performed by the Analysis and Testing Center of Sichuan University (Chengdu, China).

### 2.7. Determination of the Texture and Shear Force of Hu Sheep Muscle Tissue

The texture and shear force of Hu sheep muscle were measured using a texture analyzer (TA-XT2, Stable Micro Systems, Ltd., London, UK) and, referring to our previous study [19], using a P10 and HDP/BSW probe to measure the texture and shear force of Hu sheep muscle, respectively.

### 2.8. Determination of the Amino Acids in Hu Sheep Muscle Tissue

The amino acid content in LT muscle was measured by an L-8900 amino acid analyzer (Hitachi, Ltd., Tokyo, Japan) with reference to Wu’s study [20]. Briefly, the LT tissue was hydrolyzed by 6 M HCl for 22 h. Then, amino acid content analysis was completed using a mixed amino acid standard working solution (Anpu Experimental Technology Co., Shanghai, China), and the measurement work was finished by SICHUAN Willtest Technology Co., Ltd. (Chengdu, China).

### 2.9. Determination of the Volatile Compounds in Hu Sheep Muscle Tissue

The volatile compounds in LT muscle were measured with reference to our previous research [19]. Briefly, 3.0 g of mashed LT muscle was incubated at 80 °C for 60 min, followed by 30 min of absorbing volatiles using SPME fiber (57328-U, Supelco, Inc., Bellefonte, PA, USA). Subsequently, the volatile components of LT muscle were separated and identified using GCMS-QP2010 (Shimadzu, Co., Kyoto, Japan). All volatile compounds were identified via comparison with NIST 17 standard mass spectrometry database, and the peak area normalization method was used for the integral calculation of relative content.

### 2.10. Determination of Carbonyl and Sulfhydryl Group Content in Myofibrillar Protein

The myofibrillar protein (MP) in LT muscle was prepared using the method outlined in our previous study [19]. To determine the carbonyl content, the DNPH method was used [19,21]. Meanwhile, the sulfhydryl group content was measured using the method detailed in our earlier research [19,22].

### 2.11. Determination of TBARS in Hu Sheep Muscle Tissue

The degree of lipid oxidation in the LT tissue was determined using the TBARS method, which was outlined by Kevin [23]. In summary, 0.5 g of minced LT tissue was homogenized in 1 mL of phosphate buffer solution (PBS) containing 20 mmol/L of sodium chloride and with a pH of 6.0, using a tissue crusher (TP-24, Jelling Instrument Manufacturing Co., Tianjin, China). The homogenate was then moved to a test tube, to which 2 mL of thiobarbituric acid (TBA)-trichloroacetic acid (TCA) solution (20 mmol/L TBA in 15% (*w*/*v*) TCA) was added. The mixture was vortexed and placed in a boiling water bath for 15 min. After the sample was cooled, 2 mL chloroform was added and vortexed, then centrifuged at 3000× *g* for 15 min. Afterwards, the absorbance value of supernatant was measured at 532 nm and 600 nm.

### 2.12. Determination of Surface Hydrophobicity, Circular Dichroism, and Fluorescence Spectroscopy of MP

Briefly, the surface hydrophobicity of MP was measured by the bromophenol blue (BPB) method [24] according to our previous study [19].

Circular dichroism (CD) spectra of the MP suspension (0.4 mg/mL, dispersed in 0.6 M KCl solution) were measured with a scan in the range of 260 to 200 nm and a scan speed of 100 nm/min. Additionally, the proportion of secondary structure in MP was calculated.

Fluorescence spectra of the MP suspension (1.0 mg/mL) were measured using an F-7000 fluorescence spectrophotometer (Hitachi Inc., Tokyo, Japan) with an excitation wavelength of 295 nm and emission wavelength from 310 to 410 nm.

### 2.13. Quantitative Real-Time PCR Analysis

Total RNA was extracted from LT muscle using a commercial RNA extraction kit (Genstone Biotech, Beijing, China), following the manufacturer’s instructions. The extracted RNA was then reverse transcribed into cDNA using the TaKaRa cDNA synthesis kit (PrimeScript, Takara Biomedical Technology Co., Ltd. Beijing, China). For quantitative real-time PCR (qRT-PCR), we used TaKaRa SYBR premixes (Takara, Kyoto, Japan) and performed the reaction on a Thermo Fisher QuantStudio3 system (Thermo Fisher Scientific Co., Waltham, MA, USA). The internal reference gene was actin beta (ACTB), and the primers for qRT-PCR (see Appendix A) were synthesized by Sangon Biotech (Chengdu, China). The qRT-PCR conditions were established with reference to Zhou [25] and the 2^−ΔΔCT^ method was used to calculate the relative mRNA expression.

### 2.14. Statistical Analysis

The figures presented in this paper were generated using GraphPad Prism (version 9.0.0, GraphPad Software LLC., San Diego, CA, USA). Statistical analysis was performed using SPSS (version 26.0, SPSS Inc., Chicago, IL, USA), and significant differences were determined by analysis of variance (ANOVA) followed by the Tukey test to compare differences among treatment groups. Results are expressed as mean ± standard error, and mean values with different lowercase letters within the same parameter group indicate a significant difference (α = 0.05 level). In the charts, “Con”, “Pro”, and “Syn” represent the control group, probiotic group, and synbiotic group, respectively.

## 3. Results and Discussion

### 3.1. Growth and Slaughter Performance

Probiotics and synbiotics are widely used in livestock production to promote growth and enhance overall health by altering the intestinal tract’s ecology in a beneficial manner [7]. Probiotics are microbial feed supplements that have been successfully utilized in improving poultry health and achieving better production outcomes [7,26]. Meanwhile, prebiotics are also essential for the survival of probiotics in the gut, as they help probiotics better inhabit the digestive system and tolerate anaerobic environments [7].

As shown in Table 1, the dietary synbiotics supplementation significantly increased the body weight and body size of Hu sheep after 60 days (*p* < 0.05). Meanwhile, the daily gain and carcass weight had a tendency to increase in both the probiotics group and synbiotics group. However, probiotics and synbiotics had no effect on the slaughter rate of Hu sheep. In particular, the synbiotics compounded with *Lactiplantibacillus plantarum* N-1 and oligomeric isomaltose significantly improved the growth performance of Hu sheep. Rehman’s research [16] indicated that supplementation with probiotics (Protexin) or prebiotics (mannan oligosaccharides) could enhance broilers’ growth performance. Tagang’s study [7] also suggested that dietary feeding of yeast probiotics for 4 weeks significantly improved the growth performance of broiler chickens. Probiotics or synbiotics may contribute to the improved growth performance of Hu sheep by promoting the colonization and production of short-chain fatty acid (SCFA)-producing bacteria. This increase in SCFA production enhances the intestinal barrier function, which ultimately leads to better growth performance among newborn Hu sheep [27]. Our previous study found that *L. plantarum* N-1 is beneficial in promoting the production of intestinal SCFAs in rats [17]. Peng’s study [28] showed supplementation with *L. plantarum* B1 (2 × 10^9^ CFU/mL) significantly increased the SCFA (especially butyric acid, acetic acid, and valeric acid) content in broiler intestine after 42 days (*p* < 0.05). Furthermore, previous studies have demonstrated that supplementing probiotics in the diet of lambs can enhance feed utilization in animals [9], but the lambs were breastfed during this experiment and the daily feed intake could not be determined. In conclusion, the results of this experiment show that adding *L. plantarum* N-1 to the diet of newborn Hu sheep can enhance their growth performance, and the combination of oligomeric isomaltose and N-1 has a synergistic effect, further improving the growth performance of Hu sheep.

### 3.2. Meat Quality

Texture and color are key factors that influence consumer decisions whether to buy meat products. Table 2 shows that synbiotics improved the pH value of BB muscle 24 h after slaughter when compared to controls (*p* < 0.05). However, there was no significant effect on color, water-holding capacity, and cooking loss of Hu sheep muscle when the sheep were fed either probiotics or synbiotics. Furthermore, dietary supplementation with synbiotics increased the crude fat content and zinc in LT tissue significantly (Table 3, *p* < 0.05). As Table 4 presents, in comparison with the control, probiotics significantly increased the hardness, gumminess, and chewiness of LT muscle and decreased the springiness of BB muscle (*p* < 0.05). In addition, synbiotics significantly decreased the shear force, springiness, and resilience of LT muscle (*p* < 0.05). Nevertheless, the texture of BB muscle was not significantly affected by synbiotics.

As people’s living standards continue to improve, consumers are increasingly concerned about the quality of meat products. Tenderness, in particular, is a crucial factor in meat processing, and is also the most variable aspect of meat palatability, influenced by a variety of factors [29], such as the structure of protein [30] and fat content [31]. Additionally, intramuscular fat has a positive effect on improving meat quality, such as juiciness and tenderness of the meat [31]. Meanwhile, it is worth noting that tenderness is an important indicator for evaluating meat quality and can significantly influence consumers’ purchasing decisions. The decrease in shear force characterizes the improvement of meat tenderness, which may be due to a significant increase in crude fat content in LT muscle. For investigating the cause of fat increase, the study selected genes associated with fat synthesis for RT-qPCR analysis. The results showed that the gene expression of fatty acid synthase (FAS) was significantly upregulated in the treatment groups, while fatty acid binding protein 4 (FABP4) and lipoprotein lipase (LPL) had an upregulation trend (Figure 1). Most of the fatty acids required for fat deposition come from fatty acid synthase catalyzing the conversion of acetyl-CoA and malonyl-CoA to fatty acids, one of the key enzymes in fatty acid synthesis, which is a complex system composed of many enzymes that regulate fatty acid synthesis mainly through slow regulation [32,33]. Therefore, we suggest that the increase in fat content in LT muscle is associated with a significant upregulation of FAS gene expression. Moreover, some previous reports suggested the positive effect of probiotics on meat tenderness. Chang’s investigation [34] reported dietary supplementation of *L. plantarum* could decrease the shear force of *longissimus* muscle in pigs. In lambs too, feeding *Lactobacillus casei* HM-09 and *Lactobacillus plantarum* HM-10 reduced the shear force of the *longissimus thoracis* muscle [6]. However, the results of the study showed that the effect of probiotics and synbiotics on LT muscles was greater than that on BB muscles, which may be due to a difference in muscle fiber type between the two sites [6,35]. Furthermore, synbiotics enhanced the amount of zinc in LT muscle, which may explain why *L. plantarum* N-1 improved the absorption of zinc in the intestinal tract of Hu sheep. Zhai et al. [36] reported that *L. plantarum* CCFM242 has a superior zinc enrichment ability and could increase serum zinc levels in mice. In conclusion, our study found that the dietary addition of synbiotics improved the palatability of LT muscle in Hu sheep, and that synbiotics were more effective than probiotics. 

### 3.3. Protein Structure and Surface Hydrophobicity of MP

Circular dichroism (CD) is an important tool for assessing the secondary structure of proteins and is widely used. As shown in Figure 2a, it was observed that two negative peaks appeared at approximately 210 nm and 223 nm, which are indicative of the presence of α-helix structure in the protein [37]. Dietary supplementation with probiotics and synbiotics changed the CD spectrum of MP, which implies a change in the secondary structure of MP.

As Figure 2c presents, the fluorescence intensity of MP was reduced with dietary supplementation of probiotics and synbiotics, which indicated that probiotics and synbiotics have the potential to quench the fluorescence of MP in LT muscle and induce alterations in its tertiary structure [38]. In general, the fluorescence of a protein is primarily attributed to the presence of tyrosine residues folded in the core [37]. Therefore, the decrease in MP fluorescence intensity indicates that its internal tryptophan residues were exposed to a more hydrophilic microenvironment, which suggests a change in MP tertiary structure [39].

As Figure 2d shows, MP’s surface hydrophobicity was significantly increased with dietary supplementation of probiotics and synbiotics, which was probably the result of structural changes in MP. Studies have shown that hydrophobic amino acid residues of proteins are usually hidden inside and exposed to the surface when the protein is unfolded [29]. Therefore, increased hydrophobicity on the protein surface implies an unfolded state of the protein structure and the exposure of hydrophobic amino acids on the protein surface, which has been reported to be beneficial to improving the gel characteristics of MP [40]. Consequently, the improvement in the surface hydrophobicity of MP may have a positive impact on the quality of Hu sheep meat.

### 3.4. Amino Acids of LT Muscle

The composition of amino acids in muscle can represent the protein quality and nutritional value of meat [41]. Table 5 shows the content of 16 amino acids in LT muscle. Dietary supplementation with synbiotics significantly increased the levels of certain essential (Thr, Phe, Met) and non-essential (Asp, Ser, Tyr) amino acids in the LT tissues of Hu sheep. Meanwhile, compared with the control, the TEAA and TAA content in the synbiotics group had a significant increase (*p* < 0.05). The results of our study demonstrate that supplementing the diet with probiotics and synbiotics increased the TAA content in LT muscle significantly, which implies the nutritional quality of Hu sheep was enhanced through the increased amino acid deposition. In addition, amino acids play a crucial role in determining the taste and flavor of lamb [42]. For instance, serine and threonine contribute to its sweetness, whereas aspartic and glutamic acid provide a fresh taste [43]. Wang used metabolomics and electron emission to find that L-glutamic acid and L-aspartic acid may be important factors in increasing the umami of chicken meat [44]. Moreover, methionine is an essential amino acid that facilitates protein synthesis, DNA methylation, and polyamine synthesis in humans [45]. Research has demonstrated that the key meat flavor compounds, such as peptides and amino acids, are generated during postmortem aging through protein degradation caused by proteolytic action [46]. The study showed that the nutritional value and flavor of meat are directly related to the composition and content of amino acids, which are key evaluation indicators for meat [47]. Our results showed that the addition of synbiotics to the diet significantly increased the content of Asp, Thr, Ser, Tyr, Phe, and Met, which may be beneficial to improving the flavor and meat quality of Hu sheep.

### 3.5. Volatile Compounds in LT Muscle

The flavor of meat is a crucial factor in evaluating its quality. Figure 3 displays the principal component plot of volatile compounds in the Hu sheep LT muscle for two major factors (35.01% of PCA1, 17.3% of PCA2). A recognized lamb flavor relies on the appropriate concentration and relative proportions of multiple odor-active compounds [48]. There were 30, 30, and 31 volatile compounds, including aldehydes, alcohols, esters, alkanes, ketones, alkenes, and heterocycles, detected in the control, probiotics, and synbiotics groups of Hu sheep, respectively (Table 6). In general, aldehydes possess a relatively low odor threshold that are considered to have a crucial impact on the volatile flavor of lamb [1,48]. The aldehyde content in the control group, probiotics group, and synbiotics group was 54.35%, 54.74%, and 62.18%, respectively. The higher aldehyde content could be an increase in crude fat content in LT muscle and have a positive effect on flavor improvement. Once the lipids are broken down into free fatty acids, they become susceptible to oxidation or reaction with other Maillard reaction products, resulting in the formation of various aromatic compounds, particularly aldehydes and ketones. [49]. Moreover, the contents of 2-Pentylfuran, Octanal, (E, E)-2,4-Decadienal, etc. in the synbiotics group were higher than those in the control group, which may have an enhancing effect on the overall flavor of lamb [49]. A study by Liu et al. [50] reported that feed supplementation with *Lactobacillus plantarum* affected the expression of genes related to lipid metabolism (fatty acid synthase and stearoyl-CoA desaturase) and consequently volatile flavor compounds in Sunit sheep. In this study, it was shown that supplementation with synbiotics (*L. plantarum* N-1 and oligomeric isomaltose) also has the potential to improve the flavor of Hu sheep meat.

### 3.6. Antioxidant Capacity of Hu Sheep Tissue

Protein carbonyl content is a highly sensitive indicator for determining the extent of protein oxidation [51]. In general, when proteins are exposed to free radicals, the side chain groups of proteins tend to be converted into carbonyl groups [21]. As shown in Figure 4a, dietary supplementation with probiotics and synbiotics significantly decreased the carbonyl content of MP (*p* < 0.05). In addition, MP is reported to be rich in sulfhydryl groups, which are sensitive to reactive oxygen species and free radicals. Free radical attack on MP could promote the conversion of sulfhydryl groups to various thiol derivatives such as disulfide bonds [52]. As presented in Figure 4b, synbiotics had the potential to protect the sulfhydryl groups in MP from being destroyed. Additionally, our previous in vitro antioxidant assay results showed that *L. plantarum* N-1 had a high DPPH scavenging rate (16.31% for cell-free extracts and 31.97% for cell suspensions) [17]. Furthermore, lipid peroxidation is a widely recognized process of cellular damage in both plants and animals, and it has been utilized as an indicator of oxidative stress in various tissues and cells. The TBARS assay is a well-established method for monitoring and screening lipid peroxidation that is used to evaluate the freshness of meat [53]. As Figure 4c shows, in comparison with the control group, dietary supplementation with synbiotics inhibited lipid peroxidation and significantly decreased the TBARS level in LT tissues (*p* < 0.05). Meanwhile, the probiotics group also had the tendency to mitigate the increase in TBARS content. These findings suggest that probiotics and synbiotics supplementation enhanced the antioxidant capacity of Hu sheep muscles. Liu [6] also found that dietary probiotics supplementation improved the catalase activity and total antioxidative capacity of lamb LT muscle, which is consistent with our findings. Probiotics can colonize the gut and act as antioxidants, maintaining redox homeostasis in the gut [54], which suggests that probiotics may regulate muscle antioxidant capacity through the gut microbiota–skeletal muscle axis [6]. In conclusion, we have shown with several existing studies that the supplementation of probiotics or synbiotics to diets could improve the antioxidant capacity of muscle tissues, but the specific action mechanism needs to be further investigated.

### 3.7. Blood Lipids in Hu Sheep Serum

The serum concentrations of total cholesterol (TC) and triglycerides (TG) could be used as a crucial indicator of lipid metabolism [55]. As Figure 5 presents, dietary supplementation of probiotics significantly reduced TC and TG levels in Hu sheep serum compared with control (*p* < 0.05). However, neither probiotics nor synbiotics had a significant effect on serum HDL-C and LDL-C levels. Our previous study showed that *L. plantarum* N-1 has the potential to enhance the functional properties of lipid metabolism, which significantly decreased the serum TC and LDL-C in high-fat diet rats (*p* < 0.05) [17]. In this study, the probiotic group also tended to have reduced serum LDL-C (*p* = 0.126). Additionally, our study agrees with the results of Sudun et al. [56], who found that *L. plantarum* LLY-606 and *L. plantarum* PC-26 could significantly decrease the concentration of serum TC and TG in high-fat diet golden hamsters. However, Zhu’s study [35] showed that feeding probiotics (containing *L. plantarum* ≥ 1 × 10^8^ CFU/mL and *S. cerevisiae* ≥ 0.2 × 10^8^ CFU/mL) or synbiotics (including *L. plantarum*, *S. cerevisiae*, and xylo-oligosaccharide) to Bama mini pigs up to 125 days of age increased their serum TG and TC levels, which is contrary to the findings of our study and may be due to the use of different probiotics and prebiotics. In summary, this study showed that the addition of *L. plantarum* N-1 to the diet could enhance fat metabolism and reduce serum TC and TG in Hu sheep, which further enhanced the deposition of fat (Table 3) in the *longissimus thoracis* muscles and improved meat quality (Table 4).

## 4. Conclusions

In the present study, dietary supplementation of *Lactiplantibacillus plantarum* N-1 or its synergies with oligomeric isomaltose improved the growth performance and meat quality of Hu sheep, and oligomeric isomaltose enhanced the probiotic effect of N-1. This study contributes to our understanding of the role of *Lactiplantibacillus plantarum*-like probiotics in Hu sheep breeding. In conclusion, based on our findings, it is recommended to add N-1 to the diets of Hu sheep, and the addition of prebiotics (such as oligomeric isomaltose) as synergists is recommended to enhance the probiotic effect of N-1. Further studies are needed to gain insight into the mechanism of action of N-1 and the best application dosage in animal husbandry.

## Figures and Tables

**Figure 1 foods-12-01858-f001:**
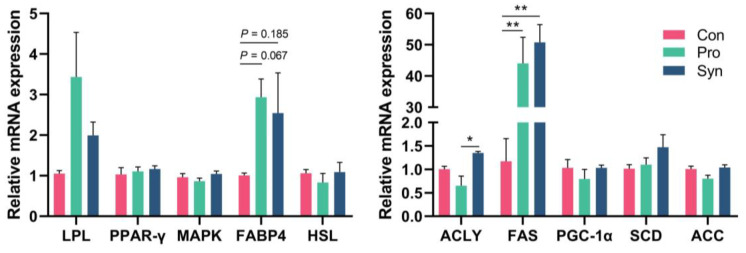
The effect of probiotics and synbiotics on mRNA expression of *longissimus thoracis* lipid synthesis-related genes (* *p* < 0.05, ** *p* < 0.01).

**Figure 2 foods-12-01858-f002:**
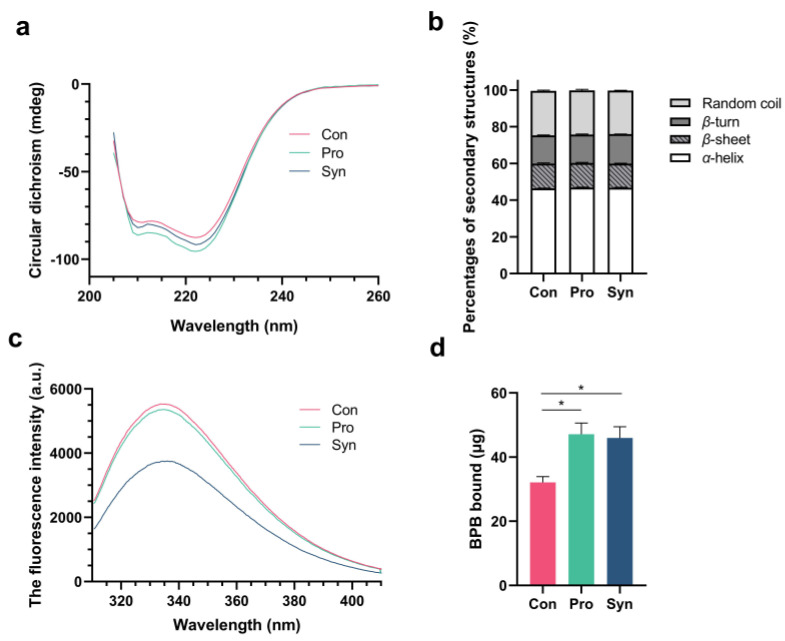
Structure and surface hydrophobicity of MP. (**a**) Circular dichroism of MP; (**b**) percentage of secondary structure of MP; (**c**) fluorescence spectra of MP; (**d**) surface hydrophobicity of MP (* *p* < 0.05).

**Figure 3 foods-12-01858-f003:**
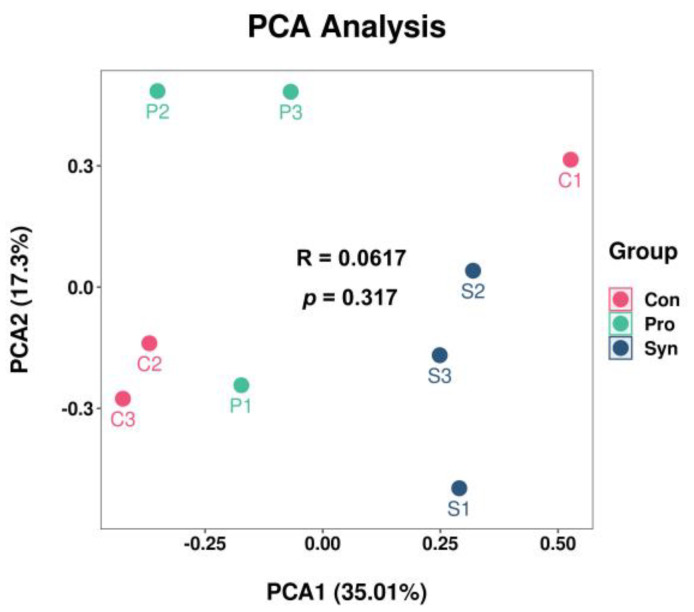
Principal component analysis (PCA) of volatile components in *longissimus thoracis*.

**Figure 4 foods-12-01858-f004:**
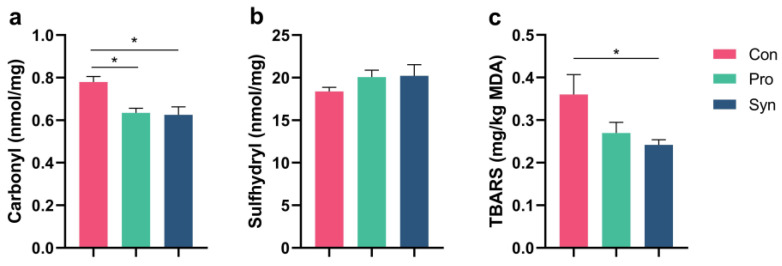
Antioxidant activity of *longissimus thoracis* muscle and MP in Hu sheep. (**a**) Carbonyl content in MP; (**b**) sulfhydryl content in MP; (**c**) TBARS content in longissimus thoracis muscle tissue (* *p* < 0.05).

**Figure 5 foods-12-01858-f005:**
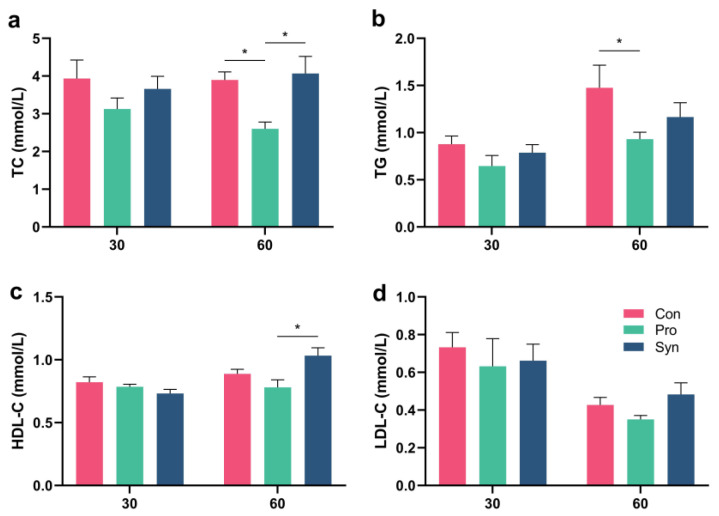
Blood lipid index of Hu sheep. (**a**) Serum total cholesterol (TC) level; (**b**) serum triglyceride (TC) levels; (**c**) serum high-density lipoprotein cholesterol (HDL-C) level; (**d**) serum low-density lipoprotein cholesterol (LDL-C) level (* *p* < 0.05).

**Table 1 foods-12-01858-t001:** Growth and slaughter performance of Hu sheep.

Parameters	Control	Probiotics	Synbiotics	SEM	*p*-Value
Initial body wight (kg)	4.36	4.83	4.95	0.200	0.479
Final body wight (kg)	13.46 ^b^	15.15 ^ab^	18.39 ^a^	0.962	0.093
Initial body length (cm)	33.50	35.00	35.00	0.748	0.683
Final body length (cm)	53.45 ^b^	55.97 ^ab^	59.20 ^a^	1.093	0.085
Initial body height (cm)	38.08	39.45	38.75	0.588	0.678
Final body height (cm)	51.90 ^b^	53.80 ^ab^	57.95 ^a^	0.874	0.003
Initial chest circumference (cm)	39.48	39.18	41.96	0.872	0.387
Final chest circumference (cm)	55.82 ^b^	58.20 ^ab^	64.00 ^a^	1.430	0.027
Average daily gain (g/d)	151.71	171.92	223.85	14.933	0.119
Carcass weight (kg)	7.18	7.60	9.86	0.542	0.079
Slaughter rate (%)	53.20	50.26	53.58	0.660	0.063

Means with different letters within the same parameter group differ significantly (α = 0.05 level).

**Table 2 foods-12-01858-t002:** The pH value, color, cooking loss, dripping loss and water-holding capacity of Hu sheep meat.

Muscle Part	Parameters	Control	Probiotics	Synbiotics	SEM	*p*-Value
*Longissimus thoracis*	pH_1h_	6.49	6.47	6.26	0.055	0.157
	pH_24h_	5.60	5.52	5.63	0.031	0.334
	*L_1h_*	45.04	42.87	43.01	0.627	0.304
	*a_1h_*	7.22 ^ab^	6.10 ^b^	8.12 ^a^	0.347	0.045
	*b_1h_*	11.07	9.99	10.44	0.237	0.180
	*L_24h_*	37.42	37.60	35.03	1.307	0.701
	*a_24h_*	5.58	4.39	5.46	0.346	0.322
	*b_24h_*	9.27	7.94	8.44	0.446	0.504
	Cooking loss (%)	41.34	42.65	41.74	0.346	0.339
	Dripping loss (%)	5.66	5.38	5.88	0.529	0.939
	Water-holding capacity (%)	95.85	95.57	95.29	0.664	0.950
*Biceps brachii*	pH_1h_	6.35	6.48	6.48	0.034	0.234
	pH_24h_	5.59	5.58	5.58	0.014	0.945
	*L_1h_*	48.75	49.19	45.27	0.851	0.114
	*a_1h_*	6.90	5.72	7.86	0.611	0.386
	*b_1h_*	11.66	12.01	11.37	0.380	0.812
	*L_24h_*	36.72	39.31	41.17	1.106	0.273
	*a_24h_*	5.32	5.48	6.70	0.385	0.300
	*b_24h_*	9.05 ^b^	10.27 ^ab^	11.11 ^a^	0.367	0.059
	Cooking loss (%)	41.82	42.38	41.95	0.577	0.931
	Dripping loss (%)	4.13	4.62	3.54	0.264	0.297
	Water-holding capacity (%)	96.95	95.78	96.97	0.350	0.303

Means with different letters within the same parameter group differ significantly (α = 0.05 level).

**Table 3 foods-12-01858-t003:** The chemical composition and mineral elements in *longissimus thoracis*.

Parameters (μg/g)	Control	Probiotics	Synbiotics	SEM	*p*-Value
K	5038.10 ^a^	4321.66 ^b^	4830.10 ^ab^	108.635	0.009
Ca	37.68	42.28	46.54	2.724	0.446
Na	631.10	660.54	559.42	21.426	0.139
Mg	246.80	248.10	256.44	3.063	0.408
Cu	1.16	1.10	1.40	0.086	0.348
Fe	13.66	16.20	13.54	1.285	0.668
Zn	25.78 ^b^	27.32 ^ab^	33.06 ^a^	1.442	0.085
Cr	ND	ND	ND		
Mn	ND	ND	ND		
Se	ND	ND	ND		
Moisture content (%)	75.30	76.15	75.78	0.347	0.443
Crude fat content (%)	3.32 ^b^	4.16 ^b^	5.00 ^a^	0.303	0.065
Crude protein content (%)	18.16	18.14	18.74	0.223	0.493
Ash content (%)	1.08	1.02	1.06	0.016	0.344

Means with different letters within the same parameter group differ significantly (α = 0.05 level, ND means not detected).

**Table 4 foods-12-01858-t004:** The texture and shear force of *longissimus thoracis* and *biceps brachii* muscle.

Muscle Part	Parameters	Control	Probiotics	Synbiotics	SEM	*p*-Value
*Longissimus thoracis*	Hardness (g)	2013.48 ^b^	3199.24 ^a^	2158.32 ^b^	142.476	<0.001
	Springiness	0.777 ^a^	0.771 ^ab^	0.721 ^b^	0.010	0.035
	Cohesiveness	0.644	0.639	0.619	0.006	0.240
	Gumminess	1422.99 ^b^	2035.86 ^a^	1340.55 ^b^	86.333	<0.001
	Chewiness	990.64 ^b^	1575.67 ^a^	982.36 ^b^	75.694	<0.001
	Resilience	0.247 ^a^	0.229 ^ab^	0.218 ^b^	0.005	0.046
	Shear force (N)	114.17 ^a^	108.38 ^a^	90.63 ^b^	4.019	0.031
*Biceps brachii*	Hardness (g)	1704.82	1891.09	1761.08	84.076	0.666
	Springiness	0.799 ^a^	0.695 ^b^	0.777 ^a^	0.011	<0.001
	Cohesiveness	0.663	0.684	0.657	0.008	0.370
	Gumminess	1121.83	1291.87	1149.06	54.398	0.405
	Chewiness	897.08	892.21	889.49	38.186	0.997
	Resilience	0.287	0.281	0.267	0.006	0.365
	Shear force (N)	122.01	124.17	114.99	3.655	0.591

Means with different letters within the same parameter group differ significantly (α = 0.05 level).

**Table 5 foods-12-01858-t005:** The amino acids content in *longissimus thoracis* muscle.

Parameters (g/100 g)	Control	Probiotics	Synbiotics	SEM	*p*-Value
Asp	1.764 ^b^	1.782 ^ab^	1.806 ^a^	0.007	0.029
Thr	0.896 ^b^	0.903 ^ab^	0.924 ^a^	0.006	0.043
Ser	0.763 ^b^	0.764 ^b^	0.794 ^a^	0.006	0.015
Glu	3.059	3.114	3.123	0.017	0.306
Gly	0.802	0.810	0.827	0.016	0.613
Ala	1.099	1.104	1.119	0.005	0.307
Val	0.929	0.938	0.946	0.003	0.100
Ile	0.867	0.887	0.888	0.004	0.046
Leu	1.562	1.572	1.592	0.006	0.070
Tyr	0.722 ^b^	0.727 ^b^	0.751 ^a^	0.005	0.016
Phe	0.820 ^b^	0.838 ^a^	0.843 ^a^	0.004	0.003
Lys	1.739	1.761	1.772	0.007	0.144
His	0.628	0.668	0.645	0.011	0.413
Arg	1.289	1.278	1.295	0.005	0.426
Pro	0.685	0.676	0.692	0.005	0.443
Met	0.521 ^b^	0.537 ^a^	0.537 ^a^	0.003	0.028
FAA	8.534	8.625	8.706	0.038	0.196
TEAA	7.333 ^b^	7.435 ^ab^	7.503 ^a^	0.030	0.039
TNEAA	10.810	10.923	11.051	0.046	0.084
TAA	18.142 ^b^	18.357 ^ab^	18.554 ^a^	0.075	0.050

TEAA: total essential amino acids, TNEAA: total non-essential amino acids, TAA: total amino acids, FAA: flavor amino acids; the FAAs included Asp, Glu, Gly, Ala, Arg and Met. Means with different letters within the same parameter group differ significantly (α = 0.05 level).

**Table 6 foods-12-01858-t006:** The volatile compounds in *longissimus thoracis* muscle.

Species	Volatile Compounds	Relative Abundance (%)			SEM	*p*-Value	Odorant Descriptors
		Con	Pro	Syn			
*Aldehydes*	Hexanal	4.41	5.82	6.19	0.488	0.332	Herbal, grassy
	2-Hexenal	ND	0.35	ND			Green apple-like, bitter almond-like
	Heptanal	2.31	3.61	2.34	0.365	0.281	Jasmine, mint, burnt fat, green
	Octanal	9.02	3.43	13.36	3.355	0.545	Citrus, floral
	(E)-2-Octenal	1.43	1.82	0.64	0.223	0.062	Fatty, wet ground, grass, coffee
	(E)-2-Decenal	1.19	1.23	0.76	0.906	0.025	Chicken fat, fried notes, waxy
	(Z)-4-Decenal	0.57	0.22	0.66	0.124	0.354	Orange, chicken fat
	(E, E)-2,4-Decadienal	0.37	0.62	0.41	0.057	0.169	Chicken fat, poultry
	(Z)-2-Nonenal	1.64	2.05	1.92	0.182	0.704	Potato peel
	Undecanal	0.10	0.19	1.73	0.567	0.482	Grassy, rain, dirt
	(E)-2-Undecenal	1.02	1.30	1.16	0.170	0.833	Waxy, fatty
	Dodecanal	9.20	8.10	6.51	1.198	0.714	Onion, green, yeast, vomit
	(E, E)-2,4-Dodecadienal	0.28	0.15	0.09	0.099	0.778	
	Hexadecanal	18.79	23.54	21.24	2.118	0.718	Sweet
	E-9-Tetradecenal	0.22	0.42	0.61	0.088	0.215	
	9-Octadecenal	0.20	ND	0.43	0.086	0.109	
	(E)-1,4-Undecadiene	0.79	0.38	0.23	0.159	0.387	
	Benzaldehyde	1.97	0.86	3.00	0.420	0.096	Almond, caramel, nutty
	4-Pentylbenzaldehyde	0.57	0.49	0.47	0.031	0.414	
	2-Methylundecane	0.27	0.16	0.43	0.100	0.600	
*Alcohols*	2-Nonen-1-ol	35.61	35.11	25.71	2.113	0.074	
*Esters*	Sulfurous acid, decyl 2-propyl ester	0.75	0.22	0.27	0.174	0.449	
	Isooctylvinyl ether	ND	0.15	0.46	0.082	0.026	
*Alkanes*	2-Methyldecane	ND	0.78	0.74	0.218	0.291	
	3-Methyltridecane	0.13	0.59	ND	0.097	0.004	
	4,8-Dimethyltridecane	1.12	3.05	0.47	1.008	0.620	
	6-Propyltridecane	0.69	5.10	4.81	1.094	0.191	
	Octadecane	ND	0.21	ND			
	2-Methyltricosane	0.38	ND	0.21	0.090	0.237	
*Ketones*	2,3-Octanedione	3.12	0.62	1.99	0.762	0.467	
*Alkenes*	1,3-Octadiene	1.88	1.13	4.09	0.913	0.444	
	2,4-Octadiene	1.38	ND	0.79	0.501	0.595	
	(Z, Z)-3,5-Octadiene	0.51	0.48	2.18	0.354	0.050	
*Heterocyclics*	2-Pentylfuran	0.20	ND	0.62	0.142	0.208	Roast, buttery

Con: control; Pro: probiotics; Syn: synbiotics. ND means not detected.

## Data Availability

The date are available from the corresponding author.

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
