# Peer review of "Effect of Dietary Supplementation of Lactiplantibacillus plantarum N-1 and Its Synergies with Oligomeric Isomaltose on the Growth Performance and Meat Quality in Hu Sheep"

_foods, 2023, doi:10.3390/foods12091858_

Round 1

Reviewer 1 Report

General comments:

The aim of the research was to investigate the influence of Lactiplantibacillus plantarum N-1 and its synbiotics with oligomeric isomaltose on the growth performance and meat quality of Hu sheep. The study is well designed and will be a valuable addition to the field. The number of lambs used in the experiment is sufficient. The applied research methods are correct. The discussion is well conducted and comprehensive. However, some changes are required in the manuscript.

The details comments are as under:

In general, it is suggested to avoid the use of words like “we”, “our” throughout the manuscript.

Introduction:

Line 51: Please provide its reference.

Line 71-72: These lines (sentence) can be deleted.

Line 73-77: It is too lengthy sentence. Please make it short for a clearer understanding.

Line 82: A reference is suggested here.

Materials and Methods:

-          Initial body weight of the male lams is missing. Please provide it.

-          The information regarding provision of feed is missing.

-          The criteria regarding provision of feed were also skipped.

Results:

Please recheck the values regarding “Ile” in Table 5. Since the values don’t have superscripts.

Conclusion:

This section can be improved. Since there is a repetition of sentences. There is a need to provide the recommendations, if any, about its application regarding Hu sheep/livestock production. 

Author Response

Dear Editors and Reviewers,

Thanks very much for the reviewers’ detailed comments on our manuscript “

Effect of Dietary Supplementation of Lactiplantibacillus plantarum N-1 and Its Synergies with Oligomeric Isomaltose on the Growth Performance and Meat Quality in Hu Sheep” (foods-2332503), which are highly valuable for revising and improving our manuscript. We have studied the comments carefully and made appropriate correction, and hope the revision meet the requirement of foods. The revised contents are highlighted in RED in the revision submitted and an item by item response is as follows:

Reviewer #1:

General comments:

The aim of the research was to investigate the influence of Lactiplantibacillus plantarum N-1 and its synbiotics with oligomeric isomaltose on the growth performance and meat quality of Hu sheep. The study is well designed and will be a valuable addition to the field. The number of lambs used in the experiment is sufficient. The applied research methods are correct. The discussion is well conducted and comprehensive. However, some changes are required in the manuscript.

The details comments are as under:

  1. In general, it is suggested to avoid the use of words like “we”, “our” throughout the manuscript.

Author: Thanks for your kind suggestion and we have made revisions in the manuscript.

  1. Line 51: Please provide its reference.

Author: Thank you for your advice and we added some references.

  1. Line 71-72: These lines (sentence) can be deleted.

Author: Thank you for your suggestion and we have deleted this sentence.

  1. Line 73-77: It is too lengthy sentence. Please make it short for a clearer understanding.

Author: Thanks for your recommendations and we have rewritten this sentence and the new sentence is “The aforementioned studies have shown the potential of probiotics or synbiotics as feed additives to improve antioxidant and meat quality in livestock and poultry. Moreover, there is increasing interest in research on their application in ruminant production as probiotics help maintain the balance of the gut microbiota and can be a possible alternative to the use of traditional antibiotics”.

  1. Line 82: A reference is suggested here.

Author: Thank you for your advice and we added some references.

  1. Materials and Methods:

- Initial body weight of the male lams is missing. Please provide it.

- The information regarding provision of feed is missing.

- The criteria regarding provision of feed were also skipped.

Author: Thank you for your careful review. We have added information on the initial weight of the lambs to the resubmitted manuscript (line 98 of resubmitted manuscript). In addition, because this experiment lasted 60 days and the lambs were breastfed throughout, no feed information was provided. Meanwhile, the ewes corresponding to the three groups consumed the same and the same amount of hay and feed daily.

  1. Results:

Please recheck the values regarding “Ile” in Table 5. Since the values don’t have superscripts.

Author: Thanks for your careful review. We examined the raw data and found that there was indeed no significant difference in Ile levels between the probiotic group (P = 0.077) and synbiotic group (P = 0.058) and the control group. And the “P-value” in Table 5 is the overall difference between the three groups and does not indicate the results of the two-way comparison. Thank you again for your thoughtful reading.

  1. Conclusion:

This section can be improved. Since there is a repetition of sentences. There is a need to provide the recommendations, if any, about its application regarding Hu sheep/livestock production.

Author: Thank you very much for your kind suggestions, we added the application of Lactiplantibacillus plantarum N-1 regarding Hu sheep/livestock production and the revised conclusion is as follows: In the present study, dietary supplementation of Lactiplantibacillus plantarum N-1 or its synergies with oligomeric isomaltose improved the growth performance and meat quality of Hu sheep and oligomeric isomaltose enhanced the probiotic effect of N-1. This study contributes to our understanding of the role of Lactiplantibacillus plantarum-like probiotics in Hu sheep breeding. In conclusion, based on our findings, it is recommended to add N-1 to the diets of Hu sheep, and the addition of prebiotics (such as oligomeric isomaltose) as synergists is recommended to enhance the probiotic effect of N-1. Further studies are needed to gain insight into the mechanism of action of N-1 and the best application dosage in animal husbandry.

Again, thank the editor and reviewers for your precious comments, which are of great help to improve our manuscript. The revision has been resubmitted to your journal, and we look forward to your positive response.

Reviewer 2 Report

The idea of this study is novel and worth trying. There are some concerns must be addressed mainly in the abstract and materials and methods. 

In the abstract , some information must be added about the treatment groups. Also, some comparisons among the treatments must be added because all information presented in the abstract were talking about the synbiotics treatment. 

line 21: how many lambs per group?

lines 25-28: please rewrite this sentence "The results ... a synergistic effect." 

lines 40-44: please provide references to support these information

lines 44-46: what do the authors mean by saying "in adequate use of antibiotics may affect meat quality?" please rewrite

lines 46-51: although these information are general, it is preferred to support it with references

line 91: what was the hypothesis of the study?

line 98: please run a power test analysis to validate whether 5 animals per treatment was enough to draw the conclusion

line 100: more information is needed about the housing whether its separate pens or group feeding? Dimensions? 

lines 101-105: how was it confirmed that the lambs consumed the assigned amounts of treatments? Since the lambs were raised with their mothers, how were the mothers prevented from consuming the water that contained the treatment. 

line 256: since lambs were raised with their mothers (they were outside the study) , how was it confirmed that any improvement that occurred to the lambs was from the treatment effect not from the mother's milk? some information should be added in the materials and methods about the mothers in terms of their milk production and feed consumption! The materials and methods in this part of the paper is not clear enough 

lines 258-259, 262-263: please delete, no need to include any numbers in the text since they were presented in the tables

lines 264-266: delete "The present .....dietary supplementation."

line 288: information must be added in the materials and methods about the methods used to evaluate body length, body height and chest circumference  and when?   

Author Response

Dear Editors and Reviewers,

Thanks very much for the reviewers’ detailed comments on our manuscript “

Effect of Dietary Supplementation of Lactiplantibacillus plantarum N-1 and Its Synergies with Oligomeric Isomaltose on the Growth Performance and Meat Quality in Hu Sheep” (foods-2332503), which are highly valuable for revising and improving our manuscript. We have studied the comments carefully and made appropriate correction, and hope the revision meet the requirement of foods. The revised contents are highlighted in RED in the revision submitted and an item by item response is as follows:

Reviewer #2:

The idea of this study is novel and worth trying. There are some concerns must be addressed mainly in the abstract and materials and methods.

In the abstract, some information must be added about the treatment groups. Also, some comparisons among the treatments must be added because all information presented in the abstract were talking about the synbiotics treatment.

  1. line 21: How many lambs per group?

Author: There were five lambs per group, and we have added comments to the abstract (line 23 of revised manuscript).

  1. lines 25-28: Please rewrite this sentence “The results ... a synergistic effect.”

Author: Thank you very much for your kind suggestions, and we have rewritten this sentence as “The results showed that dietary supplementation of N-1 tend to improve growth performance and meat quality of Hu sheep, while the synergism of N-1 by oligomeric isomaltose significantly improved their growth performance and meat quality (P < 0.05)”.

  1. lines 40-44: Please provide references to support this information.

Author: Thank you for your advice and we have added references in the manuscript.

  1. lines 44-46: What do the authors mean by saying “inadequate use of antibiotics may affect meat quality?” please rewrite.

Author: Thank you very much for your kind suggestions, and we have rewritten this sentence and replaced “inadequate” with “improper”, and the new sentence is “Meanwhile, the improper use of antibiotics in sheep farming may cause drug residues in meat products, which can be harmful to humans”.

  1. lines 46-51: Although this information are general, it is preferred to support it with references.

Author: Thank you for your advice and we have added some references.

  1. line 91: What was the hypothesis of the study?

Author: Our previous study indicated that Lactiplantibacillus plantarum N-1 has probiotic properties such as antioxidant capacity and sensitivity to antibiotics in vitro [1]. In addition, several studies have shown that some Lactobacillus plantarum could improve growth performance and meat quality in animals such as broilers and sheep [2, 3]. Therefore, we suppose that N-1 may have the potential to improve growth performance or meat quality in ruminants (sheep), especially with the help of prebiotics (prebiotics is essential for the better survival of probiotics in the intestinal tract). Consequently, in order to explore the potential probiotic effect of N-1, we hypothesised that dietary feeding of N-1 would improve growth performance and meat quality of Hu sheep, and to further enhance its improving effect, we added oligomeric isomaltose to be used in combination with N-1.

[1] Tian, L.; Liu, R.; Zhou, Z.; Xu, X.; Feng, S.; Kushmaro, A.; Marks, R.S.; Wang, D.; Sun, Q. Probiotic characteristics of Lactiplantibacillus plantarum N-1 and its cholesterol-lowering effect in hypercholesterolemic rats. Probiotics Antimicrob. Proteins 2022, 14, 337-348, doi:10.1007/s12602-021-09886-1.

[2] Liu, C.; Hou, Y.; Su, R.; Luo, Y.; Dou, L.; Yang, Z.; Yao, D.; Wang, B.; Zhao, L.; Su, L.; et al. Effect of dietary probiotics supplementation on meat quality, volatile flavor compounds, muscle fiber characteristics, and antioxidant capacity in lambs. Food Sci Nutr 2022, 10, 2646-2658, doi:10.1002/fsn3.2869.

[3] Tang, X.; Liu, X.; Liu, H. Effects of dietary probiotic (Bacillus subtilis) supplementation on carcass traits, meat quality, amino acid, and fatty acid profile of broiler chickens. Front Vet Sci 2021, 8, 767802, doi:10.3389/fvets.2021.767802.

  1. line 98: Please run a power test analysis to validate whether 5 animals per treatment was enough to draw the conclusion.

Author: We sincerely appreciate the valuable comments. We completed the power test of this experiment using R (pwr.anova.test) and the power analysis suggests that with an effect size of 1.292 (the difference between the mean body weight of Hu sheep in the synbiotics group and the mean body weight of the control group was divided by the standard deviation) and a sample size of 5 sheep per group, the power to detect a significant difference between the groups is 96.4% at a significance level of 0.05. Therefore, the sample size of 5 sheep per group is likely sufficient to draw reliable conclusions from the data. In addition, at the beginning of the experiment we planned to use more animals, but the number of lambs born in the same period in the sheep farm that met our experimental criteria was screened to have only enough for five animals per group.

  1. line 100: More information is needed about the housing whether its separate pens or group feeding? Dimensions?

Author: Thanks for your suggestion, the sheep in each group (5 lambs and 5 ewes) were fed in separate pens (pen size 4 × 4 m, fence height 0.8 m) and we have added this information in manuscript (line 101 of revised manuscript).

  1. lines 101-105: How was it confirmed that the lambs consumed the assigned amounts of treatments? Since the lambs were raised with their mothers, how were the mothers prevented from consuming the water that contained the treatment.

Author: Thank you for your careful review of the manuscript. Our treatment group was fed probiotics or synbiotics by gavage (each sheep in the treatment groups received a fixed amount of probiotics or synbiotics for the experiment, while the ewes were not exposed to probiotics or synbiotics) and we have added this information in the manuscript (line 102 of revised manuscript).

  1. line 256: Since lambs were raised with their mothers (they were outside the study), how was it confirmed that any improvement that occurred to the lambs was from the treatment effect not from the mother’s milk? some information should be added in the materials and methods about the mothers in terms of their milk production and feed consumption! The materials and methods in this part of the paper is not clear enough.

Author: Thank you for your careful review. The ewes were pre-fed under the same conditions on the sheep farm, and we treated them as non-differential animals. For the experimental grouping, we randomly divided the newborn lambs into three groups exactly according to their weight (ewes correspond to lambs one by one). In addition, all ewes were fed a fixed amount of feed, which was the same for all three groups. The milk production of the ewes, on the other hand, could not be counted. We did not intervene because the lambs fed freely on the ewes’ milk during the experiment. Overall, the lambs in the three groups were identical in all respects except that they were treated differently (fed water, probiotics and synbiotics, respectively), without any artificial intervention. Because all three groups were kept equally with the ewes, the ewes were not a variable here. The only variable is the substance that was gavaged. So we determined that the change was due to the substance we gavaged.

  1. lines 258-259, 262-263: Please delete, no need to include any numbers in the text since they were presented in the tables.

Author: Thanks to your suggestion, we have removed these sentences from the manuscript.

  1. lines 264-266: Delete “The present .....dietary supplementation.”

Author: Thanks for your suggestion, we have removed these sentences from the manuscript.

  1. line 288: Information must be added in the materials and methods about the methods used to evaluate body length, body height and chest circumference and when?

Author: Thank you very much for your kind suggestions, and we have added the specific method in the materials and methods as: During the research, the weight and body size of lambs were measured (animal weight was taken by farm staff’s holding, and the length, height and chest circumference were determined by trained sheep farm staff using measuring tape) every 15 days and serum was collected every month.

Again, thank the editor and reviewers for your precious comments, which are of great help to improve our manuscript. The revision has been resubmitted to your journal, and we look forward to your positive response.

Reviewer 3 Report

In this study, the authors examined the effect of dietary supplementation of Lactiplantibacillus plantarum N-1 with or without oligomeric isomaltose on the growth performance and meat quality of sheep. The manuscript was well-written and the obtained results were reliable. Just a few modifications should be needed, as described below.

Major

1. Discussion about amino acid

Although the authors discuss the contribution of amino acids on flavor (p. 12 line 385-395), these amino acids should be free amino acids in meat. In this study, authors measured the amino acid composition of hydrolyzed tissues. In this condition, most amino acids shown in Table 5 are utilized to compose cellular proteins in muscle. Thus authors should separate total amino acid and free amino acid in this context. Otherwise, authors can provide data on free amino acids without hydrolyzation, which are relevant to flavor.

Minor

1. mirror mouth (p.3 line 131)

I don't know what the mirror mouth means. This word should be replaced to other common word.

2. centrifuge (p.5 line 209)

Authors should use g instead of r/min in a centrifuge.

3. Figure 1

Authors should clarify what tissues they used in this experiment in the Figure legend. Authors should use "p = 0.185 and p = 0.067" instead of "0.185 and 0.067" (left panel).

Author Response

Dear Editors and Reviewers,

Thanks very much for the reviewers’ detailed comments on our manuscript “

Effect of Dietary Supplementation of Lactiplantibacillus plantarum N-1 and Its Synergies with Oligomeric Isomaltose on the Growth Performance and Meat Quality in Hu Sheep” (foods-2332503), which are highly valuable for revising and improving our manuscript. We have studied the comments carefully and made appropriate correction, and hope the revision meet the requirement of foods. The revised contents are highlighted in RED in the revision submitted and an item by item response is as follows:

Reviewer #3:

In this study, the authors examined the effect of dietary supplementation of Lactiplantibacillus plantarum N-1 with or without oligomeric isomaltose on the growth performance and meat quality of sheep. The manuscript was well-written and the obtained results were reliable. Just a few modifications should be needed, as described below.

Major

  1. Discussion about amino acid

Although the authors discuss the contribution of amino acids on flavor (p. 12 line 385-395), these amino acids should be free amino acids in meat. In this study, authors measured the amino acid composition of hydrolyzed tissues. In this condition, most amino acids shown in Table 5 are utilized to compose cellular proteins in muscle. Thus, authors should separate total amino acid and free amino acid in this context. Otherwise, authors can provide data on free amino acids without hydrolyzation, which are relevant to flavor.

Author: We sincerely appreciate the valuable comments. In this study, we measured the total amino acid content in LT muscle and did not measure the free amino acid content. We feel sorry for our carelessness and in our resubmitted manuscript, we have rewritten this part of the discussion and discussed the contribution of total amino acids on taste of meat (line 350 to 358 of revised manuscript). Thanks again for your correction.

Minor

  1. mirror mouth (p.3 line 131)

I don't know what the mirror mouth means. This word should be replaced to other common word.

Author: Thanks for your suggestion, we have revised the “mirror mouth” to “measurement probe” (line 130 of revised manuscript).

  1. centrifuge (p.5 line 209)

Authors should use g instead of r/min in a centrifuge.

Author: Thank you for your careful suggestions and we have revised in the manuscript.

  1. Figure 1

Authors should clarify what tissues they used in this experiment in the Figure legend. Authors should use “P = 0.185 and P = 0.067” instead of “0.185 and 0.067” (left panel).

Author: Thanks for your kind suggestions and we have revised the Figure 1.

Again, thank the editor and reviewers for your precious comments, which are of great help to improve our manuscript. The revision has been resubmitted to your journal, and we look forward to your positive response.

Round 2

Reviewer 2 Report

Authors have addressed all comments that were suggested on the previous version. Thank you 

Reviewer 3 Report

Authors addressed the raised issues properly in this revised manuscript.